# Influence of Acoustic Oscillations on Continuous-Flow Water Disinfection

**Anna V. Abramova [1], Vadim M. Bayazitov [1],\*, Igor S. Fedulov [1,2], Roman V. Nikonov [1], Vladimir G. Sister [2] and Giancarlo Cravotto [3]** 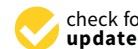

[1]   Federal State Budgetary Institution of Science, N.S. Kurnakov Institute of General Inorganic Chemistry of the Russian Academy of Sciences, GSP-1, V-71, Leninsky Prospekt 31, 119991 Moscow, Russia; abramova@physics.msu.ru (A.V.A.); if345@ya.ru (I.S.F.); thewave@yandex.ru (R.V.N.)

[2]   Federal State Budgetary Educational Institution of Higher Education, Moscow Polytechnic University, 38 Bolshaya Semenovskaya Str., 107023 Moscow, Russia; mospolytech@mospolytech.ru

[3]   Dipartimento di Scienza e Tecnologia del Farmaco, University of Turin, Via P. Giuria 9, 10125 Turin, Italy; editor.cravotto@unito.it

\*   Correspondence: vadim.bayazitov@gmail.com; Tel.: +7-925-056-0623

**Abstract:** Water disinfection and potential sterilization in continuous flow was achieved in a hybrid reactor with a broadband hydrodynamic emitter combined with ultrasonic vibrations at different frequencies and with excess pressure. Such a combination showed synergistic effects by increasing the acoustic power in the reactor vortex flow. The present combined physical treatment, compared with sonication alone, could increase microorganism inactivation by 15–20%.

**Keywords:** broadband hydrodynamic emitter; ultrasonic waveguide; hybrid technology; deionized water; disinfection; overpressure

## 1. Introduction

Disinfection is a crucial step of many technological processes for water treatment. The strategy adopted differs significantly on the basis of the quality of the water to be treated, the requirements for its disinfection and the purification degree [1]. In some cases, besides the reduction of microbiological contamination, the removal of bacteria metabolic products requires much attention [2,3]. Ion exchange technology is currently used to obtain ultrapure deionized water intended for thermal heating and power plants, as well as electronic, medical and other industries [4]. However, the use of ion exchange resins may originate new microbiological water pollution, affecting water quality. A big effort in preventive maintenance is therefore necessary to limit the risk.

Currently, there are many modern methods for water disinfection [5] characterized by positive features, but also drawbacks, such as constant regeneration and the use of complex reagents, leading to the generation of secondary byproducts and emissions. Moreover, some methods are not compatible with specific processes (e.g., the use of ion beams for deionized water). Thus, the development of new, productive, cost-effective and easily implemented disinfection methods is an urgent task [6] for deionized, potable, technical, sewage and even sea water [7,8]. In each case, the most effective method for water treatment is selected based on the level and type of chemical and biological pollution, as well as the requirements and type of use of the treated water. Typically, chemical or physical methods are used to disinfect water. While physical methods in most cases are more environmentally friendly compared with chemical methods, in the case of microbiological pollutants, they are often less efficient and more energy consuming. This is because of the need for a technological processes chain instead of a single operation. In order to avoid environmental concerns, physical methods have been carefully selected and tested.

Among physical methods, the use of an electric current as a physical agent demonstrated quite poor results. All the experiments carried out with direct current, alternating current, pulsed current and current of electrochemical activation [9] could not provide a satisfactory percentage of microorganism inactivation. Some techniques even showed a negative result. Only the formation of cold plasma in the fluid flow gave encouraging results among methods involving the use of electric current. The inactivation of microorganisms achieved by this method can reach up to 99.99%, even at very high initial concentrations ($25 \times 10^7$ bact./mL, which corresponds to wastes in medical institutions) [10]. However, being a technique not fully investigated and optimized, a real application was not reported. So far, the lack of data on the operational parameters and equipment design to guarantee efficient microorganism inactivation leaves this field unexploited.

Another promising method for water disinfection entails the use of acoustic oscillations [11] in, for example, reactors with built-in waveguide systems. However, it should be highlighted that treatment with ultrasound alone in dynamics, even with an intensity exceeding the cavitation threshold, requires a relatively long processing time. This problem can be solved by recirculation, but the performance of the equipment in this case will be reduced several times [12–15].

Aiming to intensify the process of water disinfection, in particular for the preparation of deionized water [16], the most suitable method could be the combination of low and high frequency oscillations at overpressure, which could potentially increase the mortality rate of microorganisms [17]. In this work, we investigated the influence of various combinations of physical parameters and exposure factors, aiming to develop a pilot plant. The target is a water disinfection method that meets potable water standards after the treatment of contaminated deionized water. (The limit in 1 mL is no more than 50 bacteria colony formations [18].)

## 2. Materials and Methods

On the basis of previous experience, an experimental reactor for disinfection of deionized water with the productivity 3 m$^3$/h was developed and manufactured at the Moscow Federal State Budgetary Institution of Science N.S., Kurnakov Institute of General Inorganic Chemistry of the Russian Academy of Sciences. Figure 1 shows the diagram of the acoustic equipment.

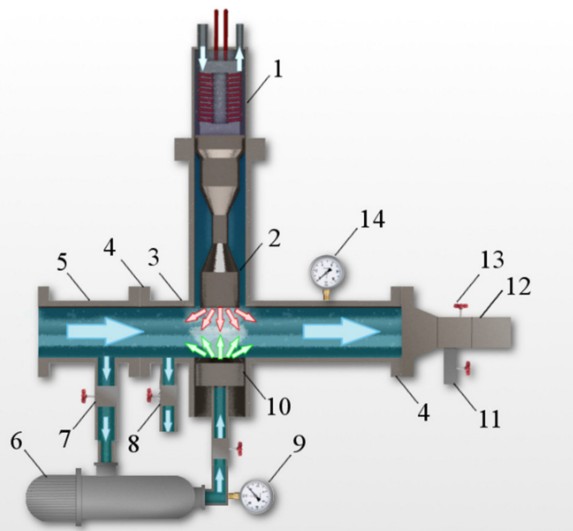

**Figure 1.** Cavitational device for disinfection of deionized water in dynamics. (1) Magnetostriction transducer, (2) waveguide emitting system, (3) disinfection chamber (reactor), (4) flanges, (5) inlet pipe, (6) pump, (7) discharge pump inlet, (8) tap for selecting water source (control), (9) pump pressure gauge, (10) hydrodynamic acoustic emitter, (11) tap for selecting water irradiated with ultrasonic waves, (12) outlet pipe, (13) tap to adjust the water flow in the reactor and (14) pressure gauge for determining the pressure in the chamber.

The basic requirement of our pilot equipment was the ability to generate the following physical phenomena in the reactor: a powerful ultrasonic field, low-frequency oscillations achieved by the hydrodynamic emitter and overpressure. All these factors can strongly affect microorganism inactivation.

The acoustic equipment operates as follows. When using only the waveguide emitting system, the initial deionized water is pumped under pressure to the inlet pipe (5), and then to the main reactor chamber (3), where it is subjected to powerful sonication with a waveguide (2). The frequency of the ultrasonic emitter is 18 kHz, and the oscillation intensity is 25 W/cm$^2$. The acoustical field is generated by a 4 kW (maximum nominal power) magnetostrictive transducer PMS15-A-18 (1), manufactured by LLC UZT (Limited Liability Company "Ultrasound Technology"), with an active surface area of 28.3 cm$^2$.

When all the equipment blocks are activated, the water is pumped through a branch (7) to the injection pump (6), and then through the hydrodynamic acoustic emitter (10) with a fundamental harmonic oscillation frequency of 2 kHz and an intensity of 3 W/cm$^2$. The fluid pressure is controlled by the pump and checked by a pressure gauge (9). The treated water is discharged through a tapering nozzle (12) connected to the flange (4), with an adjustment valve (13) to regulate the flow rate. The pressure in the reactor is determined by built-in pressure gauges (14). In order to prevent contact of the deionized water with atmospheric air, the units were sealed and checked for leaks.

Colonies of common and relatively safe *Escherichia coli* were used for biological tests.

The experiment was conducted as follows. The original deionized water contaminated with microorganisms was homogenously mixed in a tank. Then, the water was pumped to the pilot plant at 0.3 MPa. We used three different techniques for water treatment. For the first experiment, the liquid moved through the reactor, being sonicated by the waveguide emitting system. During the second experiment, a part of the initial liquid volume was divided and supplied both directly through the reactor and through the hydrodynamic emitter. During the third experiment, the liquid passed similarly to the second option, but with additional overpressure (0.05 MPa) created in the reactor chamber by the valve at the outflow. The flows, created in the pilot during the three experiments, are illustrated in Figure 2.

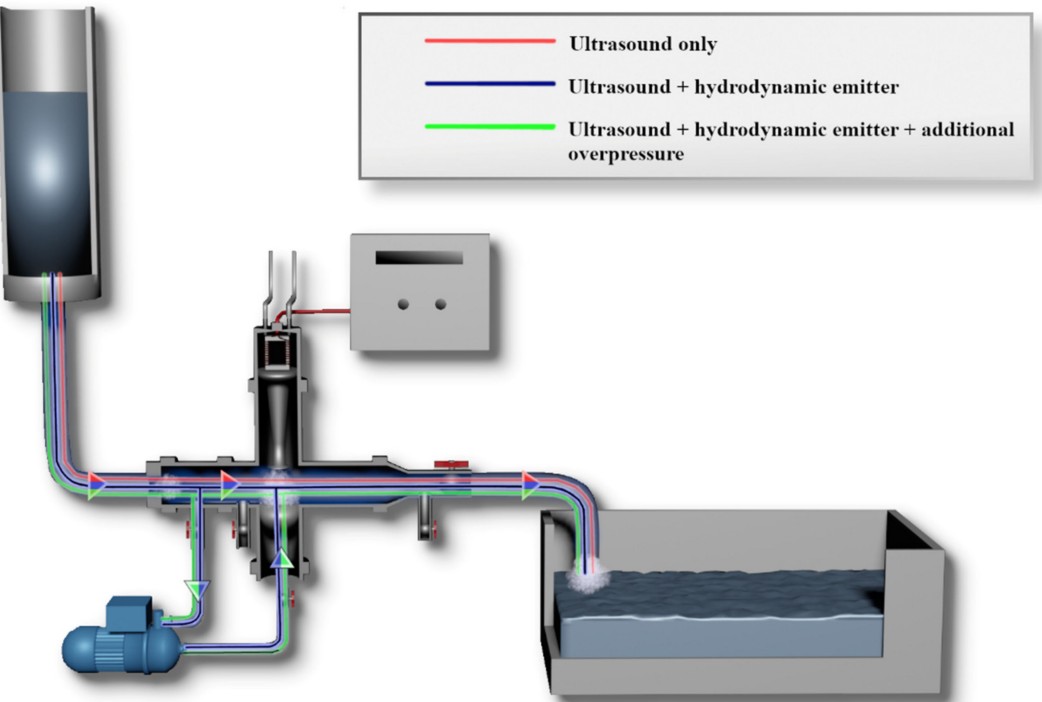

**Figure 2.** The flows of the treated liquid during the different experiments.

In each experiment, the treatment time of 1 dm$^3$ of liquid was adjusted to 0.1, 0.2, 0.3 and 0.5 s. Thus, in total, during the experimental studies, 12 dynamic fluid treatment modes were tested, which involved comparing the single and most effective physical factor (the first 4 modes), the combined effect of two factors (the second 4 modes) and the combined effect of two factors under other conditions (last 4 modes). The increase in processing time implied an assessment of the impact of the plant performance on microorganism inactivation. The treatment with only the hydrodynamic emitter or only overpressure was not studied, being known to have relatively poor effects. In the case of the hydrodynamic emitter, this is due to the fact that the intensity of the oscillations is approximately 10 times less compared with the intensity of the waveguide emitting system, while overpressure requires a significantly larger specific processing time, which cannot be created under such conditions. The combination of the hydrodynamic emitter and overpressure is also not possible because the pressure will interfere with the operation of the emitter when the cavitation threshold is approached.

Before the experiment, some liquid was taken as a control sample. The treated water after the reactor was taken at each stage of the experiment from the outlet valve (11). Between each stage of the experiment, the plant was stopped for preventive work in order to reduce data distortion. During maintenance, the equipment was tuned by the crane (13) and the regulator on the pump so that the pressure in the chamber did not differ, but the passage time of 1 dm$^3$ of test liquid increased.

The recovered samples were analyzed by a specialized microbiological laboratory, according to international standards ISO 8199:2018 Water quality—General requirements and guidance for microbiological examinations by culture.

## 3. Results

Tables 1–3 and the histogram shown in Figure 3 show the results of the experiments described above. The standard deviation of the results was close to 5%.

**Table 1.** Results of experimental studies of the effect of ultrasound alone on microorganism inactivation.

| № | Oscillation Frequency, f, kHz | Processing Time, τ, s | Intensity of Ultrasonic Oscillations, I, W/cm$^2$ | Chamber Pressure, P, MPa | Colonies Growth, C, col/mL | Inactivation, % |
|---|---|---|---|---|---|---|
| 1 | 18 | 0.1 | 25 | 0.1 | 64 | 50.8 |
| 2 | 18 | 0.2 | 25 | 0.1 | 54 | 58.5 |
| 3 | 18 | 0.3 | 25 | 0.1 | 52 | 60 |
| 4 | 18 | 0.5 | 25 | 0.1 | 48 | 63.1 |
| 5 * | - | - | - | - | 130 | - |

* Control sample.

**Table 2.** Effects of the combination of ultrasonic treatment and a hydrodynamic emitter on microorganism inactivation.

| № | Oscillation Frequency, f, kHz | Processing Time, τ, s | Intensity of Ultrasonic Oscillations, I, W/cm$^2$ | Chamber Pressure, P, MPa | Colonies Growth, C, col/mL | Inactivation, % |
|---|---|---|---|---|---|---|
| 1 | 18 and 2 | 0.1 | 28 | 0.1 | 50 | 61.5 |
| 2 | 18 and 2 | 0.2 | 28 | 0.1 | 38 | 70.8 |
| 3 | 18 and 2 | 0.3 | 28 | 0.1 | 35 | 73.1 |
| 4 | 18 and 2 | 0.5 | 28 | 0.1 | 31 | 76.2 |
| 5 * | - | - | - | - | 130 | - |

* Control sample.

**Table 3.** Results of experimental studies of the effect of the combination of ultrasonic treatment and treatment by a hydrodynamic emitter on microorganism inactivation (at overpressure).

| № | Oscillation Frequency, f, kHz | Processing Time, τ, s | Intensity of Ultrasonic Oscillations, I, W/cm$^2$ | Chamber Pressure, P, MPa | Colonies Growth, C, col/mL | Inactivation, % |
|---|---|---|---|---|---|---|
| 1 | 18 and 2 | 0.1 | 28 | 0.15 | 42 | 67.7 |
| 2 | 18 and 2 | 0.2 | 28 | 0.15 | 27 | 79.2 |
| 3 | 18 and 2 | 0.3 | 28 | 0.15 | 25 | 80.8 |
| 4 | 18 and 2 | 0.5 | 28 | 0.15 | 21 | 83.8 |
| 5 * | - | - | - | - | 130 | - |

* Control sample.

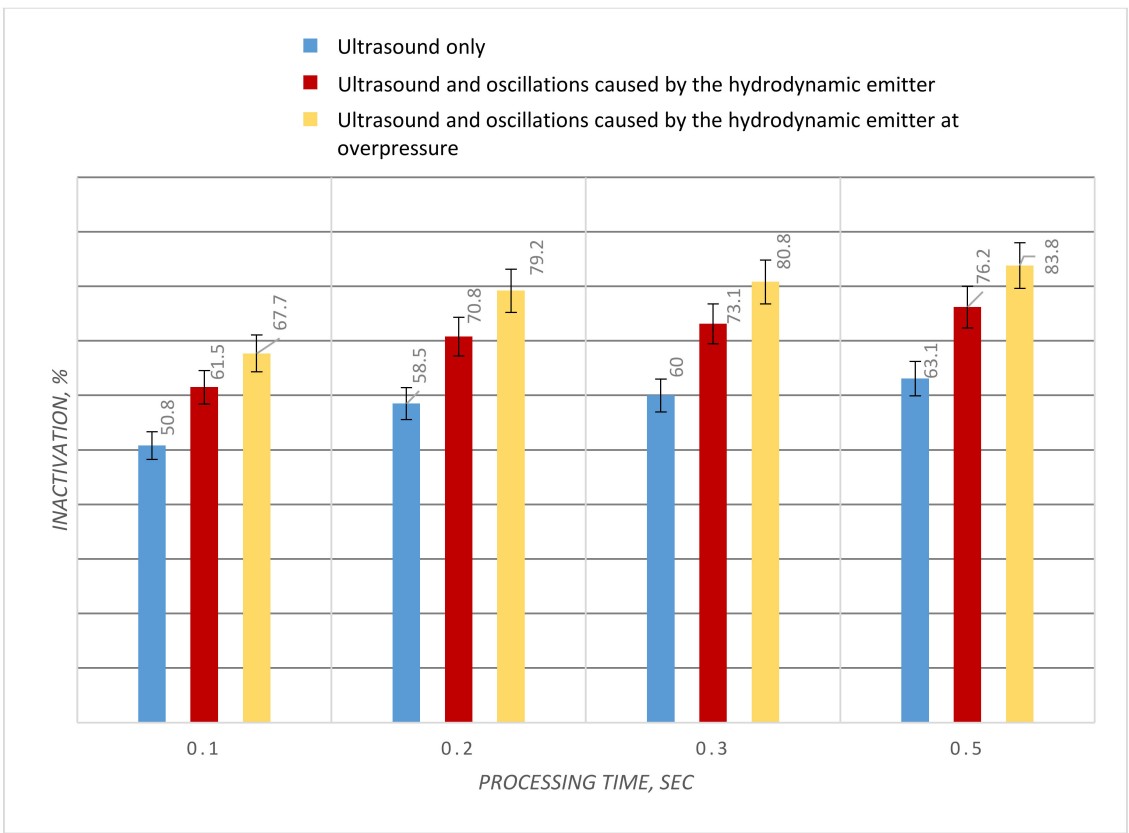

**Figure 3.** A comparative histogram of microorganism inactivation under various conditions.

## 4. Discussion

After analyzing the obtained data, it was found that disinfection of deionized water in a stream under the influence of oscillations with the frequency 18 kHz generated by an electroacoustic transducer causes less microorganism inactivation, compared with its processing when low and high frequencies of 18 kHz and 2 kHz (the main harmonic frequency of the hydrodynamic emitter) were combined. During treatment by exposure to high and low frequencies, changes in exposure (processing time) and pressure in the chamber also have a positive effect on the microorganism inactivation rate. Based on the experiments, a greater percentage of microorganism inactivation was achieved with two frequencies, with a chamber pressure of 0.15 MPa and a treatment time 1 dm$^3$-0.5 s. It means that the combination of low and high frequencies, overpressure in the chamber and, obviously, an increase in processing time collectively increase the inactivation rate. It is worth mentioning that an absolute pressure exceeding 0.5 MPa prevents the micro-shock effect of cavitation bubbles, which dramatically attenuates the inactivation (up to zero percent).

The synergistic effect can be explained by the nature of cavitation in an overpressure environment. Higher pressure values that do not compromise the cavitation significantly enhance the force of the bubbles' collapse. At the same time, the hydrodynamic emitter generates new bubbles, which are exposed to acoustic vibrations. Additional bubbles increase the overall effect on the treated medium. Another possible mechanism explaining such synergy is the homogenization of the medium in the affected area. The equalization of microbial concentrations in the treated medium allows processing without the risk of having single areas with a concentration of microorganisms that would be too high to be processed.

## 5. Conclusions

Pilot equipment for water treatment by a combination of physical phenomena (oscillations at various frequencies and pressure) was designed and manufactured. It was experimentally proven that simultaneous treatment by various frequencies increases the percentage of microorganism inactivation in flow mode processing, since the amplitude of the oscillations increases. At overpressure, the micro-shock effect of the cavitation bubbles enhances [19], and longer residence time also increases the bacteria inactivation probability. The relevant information obtained in the present work let us envision an industrial setup for water disinfection, based on simultaneous processing effects by low- and high-frequency acoustic oscillations under overpressure.

The best percentage of microorganism inactivation (~83.8%) is not yet sufficient for some industrial applications. For example, the electronics industry requires a 99.9% or complete inactivation rate [20,21]. Besides a reasonable margin of improvement, the present process could be considered as a cost-effective pretreatment. Based on the literature review, this configuration could also be combined with an electrical discharge, which causes the formation of ionized gas bubbles in a liquid. Such an approach could significantly increase microorganism inactivation to values higher than 99%.

The energy consumption of the proposed hybrid equipment is related to the application requirement. On average, 1 kW to 4 kW may be expected, of which 25% is for the pump of the hydrodynamic emitter. For example, to treat 1 $m^3$/h this way, an average of 1 kW is required. It is worth highlighting the energy saving scales up with higher productivity. In the Russian Federation, the cost of the energy would be approximately 0.01–0.05 euros (depending on the region). For comparison, treatment of 1 $m^3$ of water with chlorine costs approximately 0.005–0.025 euros, but is not fully environmentally friendly.

**Author Contributions:** Conceptualization, A.V.A.; Data curation, V.M.B. and R.V.N.; Formal analysis, V.M.B.; Investigation, I.S.F. and R.V.N.; Methodology, V.M.B. and V.G.S.; Supervision, G.C.; Validation, A.V.A.; Writing—original draft, I.S.F.; Writing—review & editing, A.V.A. and G.C. All authors have read and agreed to the published version of the manuscript.

**Funding:** This research received no external funding.

**Acknowledgments:** This work was supported by IGIC RAS state assignment.

**Conflicts of Interest:** The authors declare no conflict of interest.

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
