# Peer review of "Influence of Acoustic Oscillations on Continuous-Flow Water Disinfection"

_processes, doi:10.3390/pr8101259_

Round 1

Reviewer 1 Report

The authors reported article: Influence of Acoustic Oscillations on Continuous-Flow Water Disinfection.

The subject is very topical, it is no secret that we live in a polluted world and efforts to improve this situation are vital.

The data presented in this paper is important. The study showed that combining different methods together, such as effect of ultrasound together with hydrodynamic emitter and under overpressure conditions improves results from 63.1% to 83.8 % by microorganism inactivation.

What is the total productivity of this equipment?

Reviewer 2 Report

Please, find my comments in the attachment.

Round 2

Reviewer 2 Report

The paper can now be accepted.